# Bias in O-Information Estimation

**DOI:** 10.3390/e26100837

**Published:** 2024-09-30

**Authors:** Johanna Gehlen, Jie Li, Cillian Hourican, Stavroula Tassi, Pashupati P. Mishra, Terho Lehtimäki, Mika Kähönen, Olli Raitakari, Jos A. Bosch, Rick Quax

**Affiliations:** 1Computational Science Lab, Informatics Institute, University of Amsterdam, 1098 Amsterdam, The Netherlands; johanna.gehlen@gmail.com (J.G.); j.li4@uva.nl (J.L.); c.j.hourican@uva.nl (C.H.); 2Unit of Medical Technology and Intelligent Information Systems (MEDLAB), Department of Material Science and Engineering, University of Ioannina, 45110 Ioannina, Greece; tassistav@uoi.gr; 3Department of Mechanical and Aeronautics Engineering, University of Patras, 26504 Rio, Greece; 4Department of Clinical Chemistry, Faculty of Medicine and Health Technology, Tampere University, 33720 Tampere, Finland; pashupati.mishra@tuni.fi (P.P.M.); terho.lehtimaki@tuni.fi (T.L.); 5Finnish Cardiovascular Research Center Tampere, Faculty of Medicine and Health Technology, Tampere University, 33720 Tampere, Finland; mika.kahonen@tuni.fi; 6Department of Clinical Chemistry, Fimlab Laboratories, 33520 Tampere, Finland; 7Department of Clinical Physiology, Tampere University Hospital, 33520 Tampere, Finland; 8Centre for Population Health Research, University of Turku and Turku University Hospital, 20520 Turku, Finland; olli.raitakari@utu.fi; 9Research Centre of Applied and Preventive Cardiovascular Medicine, University of Turku, 20520 Turku, Finland; 10Department of Clinical Physiology and Nuclear Medicine, Turku University Hospital, 20520 Turku, Finland; 11InFLAMES Research Flagship, University of Turku, 20520 Turku, Finland; 12Clinical Psychology, Faculty of Social and Behavioural Sciences, University of Amsterdam, 1018 Amsterdam, The Netherlands; j.a.bosch@uva.nl; 13Institute for Advanced Study, 1012 Amsterdam, The Netherlands

**Keywords:** higher-order relationships, O-information, information synergy, bias, complex systems

## Abstract

Higher-order relationships are a central concept in the science of complex systems. A popular method of attempting to estimate the higher-order relationships of synergy and redundancy from data is through the O-information. It is an information–theoretic measure composed of Shannon entropy terms that quantifies the balance between redundancy and synergy in a system. However, bias is not yet taken into account in the estimation of the O-information of discrete variables. In this paper, we explain where this bias comes from and explore it for fully synergistic, fully redundant, and fully independent simulated systems of n=3 variables. Specifically, we explore how the sample size and number of bins affect the bias in the O-information estimation. The main finding is that the O-information of independent systems is severely biased towards synergy if the sample size is smaller than the number of jointly possible observations. This could mean that triplets identified as highly synergistic may in fact be close to independent. A bias approximation based on the Miller–Maddow method is derived for the O-information. We find that for systems of n=3 variables the bias approximation can partially correct for the bias. However, simulations of fully independent systems are still required as null models to provide a benchmark of the bias of the O-information.

## 1. Introduction

Current research in the field of neuroscience [1,2,3,4], biology [5], social science [6], and physics [7] confirms that considering higher-order relationships provides highly valuable insights into the structure and dynamics of complex systems. At the same time, the predominant practice of constructing association networks in many fields is still based on pairwise correlations, which are known to miss most or all higher-order relationships [8].

In particular, *synergy* is a measure used to quantify the higher-order relationships within a system. In a synergistic system, variables tend to be pairwise independent, while the system follows global constraints [4]. An example of a maximally synergistic system is the XOR gate, where X1 and X2 are i.i.d binary variables, and Y=X1⊕X2. This is maximally synergistic as X1 and X2 together fully determine *Y*, while still being pairwise independent of each other and of *Y* [8]. Therefore, information about any single variable in the system can only be accessed by looking at the system as a whole. As variables in synergistic systems tend to be more independent on a pairwise level, synergistic higher-order relationships are exactly the type that are elusive in a pairwise analysis. A targeted search for synergy is therefore necessary to uncover these higher-order relationships.

On the opposite side of synergy we have *redundancy*. In a redundant system, the variables are highly correlated with each other, which can be pairwise, but also at a higher level between sets of variables. An example of a system containing redundancy would be a system where variables X1 and X2 are both fully determined by a third variable, X3. In this case, both X1 and X2 would inherently carry information about each other through X3. This introduces redundancy, as each variable contains information about the remaining variables of the system.

There is currently no agreed-upon method of calculating synergy [9]. One popular framework is the Partial Information Decomposition (PID) method, introduced in 2010 by William and Beer [10], which postulates that information can be decomposed into unique, redundant, and synergistic information atoms [11]. However, no measure that satisfies the axioms, or similar sets of axioms, have been found [12] and the search continues [13]. Even if found, the calculations needed for the full decomposition may in general increase super exponentially with the size of the system [12]. Other approaches exist but have their own drawbacks. For instance, Quax et al. [8] propose a method to directly quantify synergy by using intermediate so-called synergistic random variables. However, the method has an extremely high computational complexity. Furthermore, it remains to be seen whether its concept of orthogonal decomposition actually works well for variables with a small number of states, such as binary variables. Other approaches include localized formulations [14,15] and based on information geometry [16,17].

In the absence of an agreed-upon synergy measure, an easy-to-compute heuristic that is gaining popularity is the *O-information* measure, introduced 2019 by Rosas et al. [9]. It quantifies the balance between synergy and redundancy in a system of variables; while the O-information loses the ability to quantify synergy directly, and may miss synergistic associations altogether if outweighed by redundant effects, it can nevertheless determine whether a system is “synergy-dominated” or “redundancy-dominated”. Moreover, it is calculated through a linear combination of Shannon entropy measures, making its interpretation intuitive and its calculation computationally efficient. Its computation also scales very well with the system size, easily allowing higher-order interactions of more than three variables to be analyzed [9]. These attributes motivate the use of the O-information to quantify the “synergy-dominance” within systems of variables.

Current research on the O-information highlights its potential to identify synergy- and redundancy-dominated variable groups, demonstrating its broad applicability. For example, the authors of [9,12] present proof of concept through musical score analysis. It has also been used in the neuroscience domain, as seen in [3,4]. Dynamic O-information, explored by [3], measures system evolution via synergistic interactions. Meanwhile, the authors of [18] analyze higher-order fMRI data, while [19,20] extend the O-information to the O-information rate (OIR) for studying dynamical interactions in the frequency domain, which is further applied to post-stroke brain networks [21]. The authors of [22] investigate O-information gradients in an Ising spin model and macro-economic data, while [12] introduce the local O-information for granular synergy-redundancy analysis.

However, a caveat that has been unnoticed is the statistical bias that occurs when estimating the O-information, which is introduced through the estimation of its Shannon entropy term components. As shown in this paper, these biases are exasperated by the estimation of the joint entropy, and are especially strong when the sample size is small relative to the number of bins that the variables are discretized into. This can make the interpretation of the O-information misleading, as the biases cause higher-order relationships to appear more redundancy- or synergy-dominated than they are in reality. Although one obvious way to avoid this particular type of bias is to directly compute O-information from continuous data, for which methods indeed exist [23], this is not always possible. For example, when some or all of the variables in the data may have already been discretized or are discrete by nature. No mixed method for estimating O-information exists, so if some of the data are discretized, it seems unavoidable to discretize all variables, since the continuous method will have entirely different biases as well as potentially different order of magnitude.

If O-information is to become a widely used measure to identify synergistic higher-order relationships, it is crucial to be aware of the bias in its naive estimation. With this paper, we hope to provide an intuition for the direction and magnitude of the bias in the naive O-information estimation, and how this bias behaves across different sample sizes *N*, number of discretization bins *K*, and various joint distributions for systems of n=3 variables. Moreover, an O-information bias approximation based on the Miller–Maddow entropy estimation is derived. We explore the cases in which this bias approximation does (and does not) improve the O-information estimation for n=3, and inspire future work into the bias correction of the O-information. For simplicity, our analysis focuses on these variable “triplets”, but all analyses can be extended to systems of more than n=3 variables.

## 2. Theoretical Background

### 2.1. The O-Information

The O-information, denoted by $\Omega (X^{n})$, quantifies the balance between redundancy and synergy within a system $X^{n}=\left(X_{1},\ldots ,X_{n}\right)$, and is calculated through a linear combination of Shannon’s entropy measures [9]:(1)$$\begin{array}{cc} \Omega (X^{n})\; & =\left(n-2\right)H\left(X^{n}\right)+\sum_{j=1}^{n}\left[H\left(X_{j}\right)-H\left(X_{-j}^{n}\right)\right], \end{array}$$
where $H(X_{j})$ is the entropy of variable $X_{j}$, $H(X^{n})$ is the joint entropy of the system $X^{n}$, and $H(X_{-j}^{n})$ is the joint entropy of the system $X_{-j}^{n}=\left(X_{1},\ldots ,X_{j-1},X_{j+1},\ldots ,X_{n}\right)$ without variable $X_{j}$. If the O-information $\Omega (X^{n})>0$, this implies the system is redundancy dominated; meanwhile, if $\Omega (X^{n})<0$, this implies synergy domination. If the O-information is zero, the variables may be fully independent, or the synergy and redundancy in the system cancel each other out perfectly [9].

### 2.2. Properties of the O-Information

The O-information is a symmetric measure [9], and does not require the specification of a target variable. It is used on cross-sectional data, but extensions of the O-information to time-series data have been explored in the previous literature [22].

Tight bounds on the O-information are provided in [9]. Let $K_{j}$ be the cardinality of the discrete random variable $X_{j}$. Then, let $K_{\max}=\max\left\{K_{1},\ldots ,K_{j}\right\}$ be the cardinality of the variable with the largest cardinality in system $X^{n}$. The upper and lower bounds on the O-information are then:(2)$$\left(2-n\right)\log K_{\max}\leq \Omega \left(X^{n}\right)\leq \left(n-2\right)\log K_{\max},$$
where the log is always calculated with the base of 2 unless specified otherwise. The authors of [9] further provides proof that when all variables are discretized into the same number of bins $K_{j}=K\forall j$, the maximum O-information $\Omega \left(X^{n}\right)=\left(n-2\right)\log K$ can be achieved by a system if—and only if—the outcome of each random variable can be inferred by any other variable in the system. Then, the system is maximally redundant. The minimum O-information $\Omega \left(X^{n}\right)=\left(2-n\right)\log K$ can be achieved if—and only if—$X^{n-1}$ are independent and uniformly distributed, and $X_{n}=\sum_{j=1}^{n-1}X_{j}\;\mathrm{mod}\;\left(K\right)$ [9]. Then, the system is maximally synergistic.

### 2.3. O-Information Estimation

When the underlying marginal and joint probability distributions are not known, like in most empirical datasets, the O-information must be estimated from the available data. Let the estimation of O-information and its entropy terms be denoted as:(3)$$\begin{array}{cc} \hat{\Omega }\left(X^{n}\right)\; & :=\left(n-2\right)\hat{H}\left(X^{n}\right)+\sum_{j=1}^{n}\left[\hat{H}\left(X_{j}\right)-\hat{H}\left(X_{-j}^{n}\right)\right]. \end{array}$$

It is well-known that a limited sample size results in a downward bias in the estimation of the entropy terms H(X), $H(X^{n})$ and $H(X_{-j}^{n})$. This can be proven using Jensen’s Inequality, but can also be intuitively explained: with a finite sample size we observe rare events less, or maybe not at all. With the naive entropy estimation, we then assign these events a smaller probability than they actually have. Therefore, the probability distribution seems less spread out, and by definition this results in a smaller entropy estimation. It is then clear to see that these biases may subsequently introduce a bias into the estimation of O-information (Equation 3), such that
(4)$$E\left[\hat{\Omega }\left(X^{n}\right)\right]=\Omega \left(X^{n}\right)-\delta ,$$
where δ is the bias in O-information estimation. The balance of the biases introduced by each of the entropy terms in (Equation 3) determines whether O-information is under- or overestimated. The marginal and joint distributions of the variables in the system will ultimately determine the sign of O-information bias. If the bias is not accounted for, a sample size that is too small may result in a system of variables $X^{n}$ to appear to be more redundancy or synergy-dominated than it is in reality.

## 3. Previous Literature

### 3.1. Entropy Bias Approximation Techniques

The bias in the naive entropy estimation has been explored extensively in the previous literature. The most prominent methods include the Miller–Maddow bias approximation [24], the Jackknife entropy estimation [25], the Grassberger estimation [26], and several Bayesian methods initiated by Wolpert and Wolf [27]. In this paper, we extend the Miller–Maddow method from the entropy to O-information due to its low computational cost in comparison to the Jackknife method, its simplicity in comparison to the Grassberger method, and its accuracy in comparison so the Bayesian methods.

#### The Miller–Maddow Entropy Bias Approximation

The Miller–Maddow method [24] estimates the expected value of the naive entropy estimations $E[\hat{H}\left(X\right)]$ by making a Taylor approximation of the function $-\hat{p_{i}}log\left(\hat{p_{i}}\right)$ around each bin’s respective probability $p_{i}$. This is repeated for all possible outcomes of $\hat{p_{i}}$, and the expected value over all Taylor approximations per bin *i* is taken. See Appendix A for the full derivation.

A requirement of the Miller–Maddow method is that *N* is in the asymptotic sampling regime, which means that the sample size is large enough such that each outcome of the random variable is observed many times [28].

When taking the Taylor approximation to the second order, the resulting Miller–Maddow entropy estimator is
(5)$$E{\left[\hat{H}\left(X\right)\right]}_{MM}=H\left(X\right)-\frac{K-1}{2Nln(2)}.$$

Let the Miller–Maddow bias approximation term of entropy be denoted by $\delta_{H}$, such that
(6)$$\delta_{H}:=\frac{K-1}{2Nln(2)}.$$

This entropy bias approximation (Equation 6) only depends on the number of bins, *K*, and the number of observations, *N*. We see that as N→∞, the bias approximation $\delta_{H}\to 0$. This makes sense intuitively, as the naive plugin estimator $\hat{p_{i}}\to p_{i}$ as N→∞. Moreover, as K→∞, the bias approximation $\delta_{H}\to \infty$, as increasing the number of bins decreases the mean number of possible observations per bin.

### 3.2. Mutual Information Bias Approximation Techniques

Beyond the bias approximations for entropy estimations, several bias approximation methods have been explored for the estimation of other information theoretic measures as well, such as the mutual information.

In this paper, however, we derive the O-information bias approximation from the Miller–Maddow entropy bias approximation. As the mutual information is a composite measure of various entropy terms, each with their own biases, it would be more difficult to attribute the behavior of the O-information estimation bias to its individual mutual information components. The entropy formulation is preferred over a mutual information formulation because it is clearer to separate the bias of each entropy term in the O-information estimation (Equation 3). Especially as we investigate the O-information estimation bias for various joint distributions which each have distinct behaviors of $H(X^{n})$ and $H(X_{-j}^{n})$, it is important to keep the sources of the bias in each of these terms as clear as possible. Moreover, it is more convenient to generalize the bias of the joint entropy of a system with n=3 variables to systems of any other number of variables. This makes the entropy formulation of the O-information estimation more compatible when considering the extension of this analysis to various system sizes in future work.

### 3.3. Permutation Tests

Further, we do not use permutation tests to quantify the bias of O-information. For a single triplet, the permutation test involves the permutation of data for each variable, forcing independence between the variables, and then calculating the new O-information from the resulting joint probability distributions. This process is repeated many times per triplet, and finally the original O-information estimation is compared to the distribution of the O-information estimations of the permuted data. As the permutation destroys any relationships between the three variables, the mean O-information after permutation should, in theory, be zero. If the mean is not zero, the deviation from zero represents the bias in the O-information estimation that persists. However, since this process is repeated for each triplet individually, estimating the bias using permutation tests is computationally very expensive. This is particularly true when trying to identify synergistic triplets in a larger dataset. Therefore, we develop a bias approximation in this paper that can be reasonably applied to the entire dataset, making the process much more convenient for exploratory analyses.

## 4. Methods

### 4.1. Derivation of the Miller–Maddow O-Information Bias Approximation

The Miller–Maddow entropy estimation (Equation 5) applies to the entropy of a single random variable. In order to approximate the bias in the O-information estimation, we must extend the entropy estimations to the joint entropy terms $\hat{H}\left(X^{n}\right)$ and $\hat{H}\left(X_{-j}^{n}\right)$.

The Taylor expansion process is the same for the joint entropy estimation $\hat{H}\left(X^{n}\right)$ as for the entropy estimation $\hat{H}\left(X\right)$, just that we are now looking at the probabilities of possible bin *combinations*, rather than the probabilities of individual variables’ bins. The estimated joint entropy therefore becomes
(7)$$E\left[\hat{H}\left(X^{n}\right)\right]\approx H\left(X^{n}\right)-\left(\frac{K^{\left(n\right)}-1}{2Nln(2)}\right),$$
where $K^{\left(n\right)}$ is the number of possible bin combinations the *n* variables in the system can have. We see that this yields the following joint entropy bias approximation term:(8)$$\delta_{Joint}:=\frac{K^{\left(n\right)}-1}{2Nln(2)}.$$

Similarly, we can derive that the estimation bias of the joint entropy of system $\hat{H}\left(X_{-j}^{n}\right)$ can be approximated by:(9)$$\delta_{Joint}^{-j}:=\frac{K^{\left(-j\right)}-1}{2N\ln(2)},$$
where $K^{\left(-j\right)}$ is the number of possible bin combinations of all variables in the system except for variable $X_{j}$.

As the estimation $\hat{\Omega }\left(X^{n}\right)$ is a random variable, we can take the expected value of (Equation 3):(10)$$E\left[\hat{\Omega }\left(X^{n}\right)\right]=\left(n-2\right)E\left[\hat{H}\left(X^{n}\right)\right]+\sum_{j=1}^{n}\left[E\left[\hat{H}\left(X_{j}\right)\right]-E\left[\hat{H}\left(X_{-j}^{n}\right)\right]\right].$$

Substituting the bias approximations (Equation 6), (Equation 8), and (Equation 9) and simplifying we obtain the final O-information estimation derived from the Miller–Maddow method:(11)$$E\left[\hat{\Omega }\left(X^{n}\right)\right]\approx \Omega \left(X^{n}\right)-\frac{1}{2N\ln(2)}\left[\sum_{j=1}^{n}\left(K_{j}-K^{\left(-j\right)}\right)+\left(n-2\right)\left(K^{\left(n\right)}-1\right)\right].$$

Thus, we find that by extending the Miller–Maddow entropy estimation to the joint and conditional entropy, the O-information bias can be approximated by the bias term:(12)$$\delta_{MM}:=\frac{1}{2N\ln(2)}\left[\sum_{j=1}^{n}\left(K_{j}-K^{\left(-j\right)}\right)+\left(n-2\right)K^{\left(n\right)}-n+2\right].$$

### 4.2. Simulating the Bias in the Naive O-Information Estimation

The effect of sample size, *N*, and number of bins, *K*, on the bias of the naive O-information estimation can be investigated in a controlled environment through simulations. In these simulations, we construct variable triplets with joint distributions for which we *know* the true $\Omega (X^{n})$, and compare this to our estimation $\hat{\Omega }\left(X^{n}\right)$ that is based on solely the available, simulated samples.

#### 4.2.1. The Discretization Strategy

As we explore the discrete case in this paper, the sampled variables must be discretized before estimating their O-information. In empirical data, the true underlying bin widths are not known, and the best we can do is to construct the bins based on the available sample we observe. In our case, we use the quantile binning strategy, where the bin width is determined such that the number of observation in each bin stays the same. For instance, with K=10, the lowest 10% of observations are allocated into bin 1, the highest 10% of observations into bin 10. This results in the marginal distributions of each variable to be approximately uniform.

Of course, in our simulations, we do know the true underlying marginal and joint probability distributions of the variables. However, as we want our simulations to reflect the scenario of working with real empirical data, we construct the bins for each simulated variable according to the observed outcomes we have sampled, rather than the true underlying distribution, while this likely reduces the accuracy of the bias correction in these simulations, it allows us to obtain an idea of the performance of the Miller–Maddow bias correction on real, empirical data.

#### 4.2.2. The Simulated Distributions

We construct three different systems: The first system achieves the theoretical maximum O-information (a fully redundant system), the second achieves a theoretical O-information of zero (a system of fully independent variables), and the third achieves the theoretical minimum O-information (a fully synergistic system).

These three distributions are constructed as follows:Redundant triplet: each variable is a copy of each other $X_{1}=X_{2}=X_{3}$. Each variable is uniformly distributed and discretized into *K* bins. This achieves the maximum possible O-information of a triplet with *K* bins: Ω=log(K) [9].Independent triplet: each variable is independent and uniformly distributed, discretized into *K* bins. Due to the variables’ independence, the true O-information Ω=0 regardless of *N* and *K*.Synergistic triplet: $X_{1}$ and $X_{2}$ are independent and uniformly distributed, discretized into *K* bins. $X_{3}=X_{1}+X_{2}\;\mathrm{mod}\;K$. This achieves the minimum possible O-information, Ω=−log(K) [9].

We set the number of bins to be equal across the three variables in the system, such that $K_{1}=K_{2}=K_{3}=K$. This allows us to apply the same bias approximation method across all triplets in the dataset.

It is important to realize that these three systems require a different minimum sample size in order to theoretically be able to estimate their joint probability distributions. This is simple to see for a system of independent variables: the number of samples needed increases exponentially with the number of variables in the system *n*. In contrast, for a system of fully redundant variables, we only need a minimum sample size of *K* to be able to estimate the joint probability distribution, regardless of the number of variables in the system. In a fully synergistic system, all variables are independent from each other, except for the last, which is fully determined by all others. Thus, the theoretical minimum sample size needed to estimate the joint distribution of all variables except for the last increases exponentially (just like in the fully independent system). To estimate the joint distribution of the entire system however, only $K^{n-1}$ samples are needed.

#### 4.2.3. Simulation Setup

The tested number of bins ranges from K=2 to K=50, increasing in intervals of 1. The tested sample sizes range from N=500 to N=20,000, increasing in intervals of 500. 30 trials are carried out due to the stochasticity in the data generation, leading to 30 naive O-information estimations per (N,K) combination per system:$${\hat{\Omega }}_{1},\ldots ,{\hat{\Omega }}_{30}.$$

Let the mean naive O-information estimation over the 30 trials be denoted as $\bar{\hat{\Omega }}$, and the resulting mean bias be denoted as
(13)$$\delta :=\Omega -\bar{\hat{\Omega }}.$$

As the data are generated artificially, we know the underlying number of possible bin combinations $K^{\left(-j\right)}$ and $K^{\left(n\right)}$. Note that *K* is known by definition of the quantile binning strategy. These values can then be substituted into the equation for the O-information bias correction (Equation 12). Again, we obtain 30 bias-corrected O-information estimations
$${\hat{\Omega }}_{BC1},\ldots ,{\hat{\Omega }}_{BC30},$$
and we let the mean observed bias be denoted as
(14)$$\delta_{BC}:=\Omega -\bar{{\hat{\Omega }}_{BC}}.$$

However, in real empirical data, the true $K^{\left(-j\right)}$ and $K^{\left(n\right)}$ values are usually not known, and have to be estimated from the available data. We estimate them here by simply counting the number of joint bin combinations that occur in the data. Let the mean observed bias using the estimated $K^{\left(-j\right)}$ and $K^{\left(n\right)}$ values be denoted as
(15)$$\delta_{BC^{\prime }}:=\Omega -\bar{{\hat{\Omega }}_{BC^{\prime }}}$$

The mean error of the naive O-information estimation can then be compared to the mean error of the bias-corrected O-information,
(16)$$\varepsilon^{\prime }:=|\delta_{BC^{\prime }}|-|\delta |.$$

We focus on the bias approximation using $\delta_{BC^{\prime }}$ rather than $\delta_{BC}$, as we usually do not know the true $K^{\left(-j\right)}$ and $K^{\left(n\right)}$ values and must estimate them from the available data.

### 4.3. Application to the Young Finns Study

Next, we aim to investigate how our knowledge of the bias in the O-information estimation affects the conclusions we draw about the higher-order interactions of variable triplets in an empirical dataset. This is explored on the empirical dataset of the Young Finns Study (YFS), which is one the largest and long-standing observational cohort studies globally that has followed a cohort of individuals from childhood to adulthood [29]. Specifically, we are interested whether we can discover synergistic higher-order relationships between cardiovascular disease (CVD) and depression variables. The subset of data we use is the 2007 wave of the YFS, and contains lipidomic data, metabolomic data, depression scores, and CVD related phenotypes. For more information on the YFS dataset, see Appendix B. Only the participants who appear in all four datasets are kept in our final dataset, ensuring that lipidome-, NMR-, depression-, and CVD-related phenotype data are available for all participants. This leaves a total of *N* = 1684 participants. Any missing values are imputed with the median of the feature.

The dataset contains a mix of continuous and discrete variables. Variables are discretized using the quantile K-bins method, where all variables are discretized into K=10 bins with an equal number of samples in each bin. Just like in our simulations, this results in the marginal probability distributions of the variables being approximately uniform after discretization, and they are perfectly uniform if *N* is a multiple of *K*. As the number of bins, *K*, is one of the main components in the O-information estimation and the bias approximation, $\delta_{MM}$, it is crucial that all variables are discretized into the same number of bins, *K*. The quantile binning strategy ensures that this is the case, given that N≥K. An equal-width binning does not guarantee this, as empty bins can remain, even if N≥K.

After a feature selection described in Appendix C, we compute the naive and bias-corrected O-information estimations for all possible sets of variable triplets. The resulting distribution of O-information estimations from the empirical data are then compared to the O-information estimations of our simulations with the same *N* and *K* values.

## 5. Experiments and Results

### 5.1. Bias of the Naive O-Information Estimation in Simulations

First, we explore the effect of varying the sample size, *N*, and the number of bins, *K*, on the mean naive O-information estimation $\bar{\hat{\Omega }}$ and its bias δ (Equation 13). Figure 1 shows $\bar{\hat{\Omega }}$ for all simulated distributions, and Figure 2 shows δ for all simulated distributions. Note that the bias of the redundant triplet is shown for a range of much smaller sample sizes, with *N* only going up to N=50. As the joint entropy of the fully redundant triplet is equal to the marginal entropies, the O-information bias is much smaller; thus, this is the scale needed to observe the bias for up to K=50 bins.

Figure 2 can be read as follows: if δ>0 (red), then $\bar{\hat{\Omega }}$ is more synergistic than the true Ω. If δ<0 (blue), then $\bar{\hat{\Omega }}$ is more redundant than Ω. In this figure, we also plot the N=K line for the redundant triplet, the $N=K^{2}$ and $N=K^{3}$ lines for the independent triplet, and the $N=K^{2}$ line for the synergistic triplet. As explained in Section 4.2.2, these boundary lines are the theoretical minimum sample sizes that are needed to observe all possible joint bin combinations.

The bias of the naive O-information estimation, δ, is particularly strong for the triplet of independent random variables, being strongly biased towards synergy as seen in Figure 2. The most severe bias occurs approximately in the space where $K^{2}≲N≲K^{3}$. This is demonstrated through the red shaded area between the dashed lines of $N=K^{2}$ and $N=K^{3}$. Interestingly, despite the lower N/K ratio, a decreased (but still positive) bias is observed in the region where $N≲K^{2}$, demonstrated by the lighter shades below the $N=K^{2}$ line.

For the redundant triplet, we observe that $\bar{\hat{\Omega }}$ is biased strongly towards synergy, being most severe when N/K is the smallest. Around N=K, however, the bias fades to zero. For the synergistic triplet, we observe that the $N=K^{2}$ line approximately traces the boundary above which $\bar{\hat{\Omega }}$ is biased. The bias δ has around the same magnitude as in the redundant and independent triplet, but an opposite direction, while the redundant and independent triplets were biased towards synergy, the synergistic triplet is biased towards redundancy.

Looking at the standard deviations of the naive O-information estimations on Figure 3, we observe that they are quite low for all three simulated systems, ranging up to a standard deviation of 0.08. The lowest standard deviations are seen for the redundant triplet, which is approximately zero for all (N,K) combinations. The standard deviations of the synergistic and independent triplets are comparatively higher, especially for small *N*.

### 5.2. The Miller–Maddow O-Information Bias Approximation

Next, we explore the accuracy of the O-information bias approximation $\delta_{MM}$ derived in (Equation 12) on our simulations. Recall that $K^{\left(-j\right)}$ and $K^{\left(n\right)}$ are estimated from the data by simply counting the number of unique joint bin occurrences. We are mainly interested in the decrease in the estimation error after the bias correction, which is given by $\varepsilon^{\prime }$ in (Equation 16).

Figure 4 shows $\varepsilon^{\prime }$ for all simulated (N,K) combinations. A negative value (blue) implies that the bias-corrected O-information estimation $\bar{{\hat{\Omega }}_{BC^{\prime }}}$ is more accurate than the naive O-information estimation $\bar{\hat{\Omega }}$. A positive value (red) implies that the naive estimation is more accurate. This analysis highlights the situations in which the bias approximation $\delta_{MM}$ (Equation 12) works well, and in which situations it fails.

To better visualize the behavior of $\bar{\hat{\Omega }}$, $\bar{{\hat{\Omega }}_{BC^{\prime }}}$, and the true O-information Ω, Figure 5 shows these values for the K=10 slice and Figure 6 for the K=30 slice of the heatmap in Figure 4. These two *K* values are chosen because they exhibit quite different behaviors, which provides an interesting contrast.

For the independent triplet, four separate sections can be identified in Figure 4. There is close to no bias in the naive estimation when $N≳K^{3}$, and the bias correction is also approximately zero in this case. In Figure 5, this is equivalent to the section of large *N* where both $\bar{\Omega }$ and $\bar{{\hat{\Omega }}_{BC}}$ have almost converged to the true O-information Ω=0. When $K^{2}≲N≲K^{3}$, the bias correction indeed reduces the bias, as indicated by the negative $\varepsilon^{\prime }$. In Figure 6 this is the section in which both $\bar{\hat{\Omega }}$ and $\bar{{\hat{\Omega }}_{BC}}$ approach Ω, but $\bar{{\hat{\Omega }}_{BC}}$ approaches it faster. Then, as *N* decreases further, there is a small section where the naive estimation and the bias-corrected estimation have approximately equal errors. In Figure 6, this is equivalent to the part where the two O-information values intersect. This only happens when the number of bins K≳15, so we do not observe this intersection in Figure 5. For small *N* and large *K*, $\varepsilon^{\prime }>0$, implying that the naive estimation is more accurate than the bias-corrected estimation. This is also reflected in Figure 6 for small sample sizes, where the orange line, $\bar{{\hat{\Omega }}_{BC^{\prime }}}$, briefly dips below the blue line, $\bar{\hat{\Omega }}$.

For the redundant triplet we also observe an interesting pattern. The bias correction seems to improve the accuracy of the O-information estimation for most N≲K. On Figure 5 and Figure 6, this is reflected in the part where both $\bar{\hat{\Omega }}$ and $\bar{{\hat{\Omega }}_{BC^{\prime }}}$ approach Ω, when N<K, but $\bar{{\hat{\Omega }}_{BC^{\prime }}}$ approaches it faster. On and around the N=K line, the naive estimation becomes more accurate than the bias-corrected estimation. In Figure 5 and Figure 6, this is reflected by $\bar{\hat{\Omega }}$, intersecting with the true Ω, while the $\bar{{\hat{\Omega }}_{BC^{\prime }}}$ overshoots the true Ω value at N=K. Gradually, however, $\bar{{\hat{\Omega }}_{BC^{\prime }}}$ approaches the true value again as *N* increases further. There is also an intriguing pattern of white dots along the N=K line in the heatmap in Figure 4. For the larger sample sizes the error fades to approximately zero.

For the synergistic triplet, the pattern is much more straightforward. The bias correction improves the accuracy of the O-information estimation wherever there is a bias in the naive estimation to begin with, which is where $N≳K^{2}$. If $N≲K^{2}$, the bias approximation is also around zero.

### 5.3. Application to the Young Finns Study

Applying this exploration to an empirical dataset, the O-information of all possible triplets in the feature selected YFS dataset is estimated. Figure 7 compares results from the empirical YFS data with the results from our simulations. For the empirical data, it shows the distribution of the naive O-information estimations and the estimations with the bias correction applied (Equation 12). For the simulations, dashed vertical lines at $\bar{\hat{\Omega }}=-0.377$ and $\bar{{\hat{\Omega }}_{BC^{\prime }}}=-0.142$ show the mean naive O-information estimation and the mean bias-corrected O-information estimation of the simulated independent triplets with N=1684 and K=10. The dark shaded distributions around these dashed lines additionally show the distribution of these O-information estimations over all 30 trials.

Taking a closer look at the 20 most synergistic triplets in the YFS dataset, we note two important results. Firstly, 90% of these highly synergistic triplets contain ratio variables. For example, a ratio triplet might look like “MHDLC, MHDLTG, MHDLTGPCT”, where MHDLC is the total cholesterol in medium HDL, MHDLTG are the triglycerides in medium HDL, and MHDLTGPCT is the ratio of the triglycerides to total lipids ratio in medium HDL. Note that lipids include both cholesterol and triglycerides (as well as other types of lipids) [30]; thus, the MHDLTGPCT variable contains information about the ratio of cholesterol to triglycerides. This provides a proof of concept of the O-information in a technical sense, as these ratios embody the core concept of synergy: after all, a ratio can only be computed if the two input variables are known. A table of the 20 most synergistic triplets with indication which ones contain such a ratio variable can be found in Appendix D.

However, in a medical context, the synergy of ratio triplets is not relevant. Therefore, we exclude such triplets from our further analysis. We additionally remove variables that are sums or means of other variables. Triplets containing these composite variables are likely to appear highly synergistic due to the inherent mathematical relationships between each other. Consequently, these triplets are not medically relevant as they do not represent actual synergistic biological pathways.

On this remaining dataset, we now explore whether the bias correction gives any further insight on the synergistic triplets. Specifically, whether correcting for the bias changes the order of synergistic triplets in the dataset, or if it reveals any new synergistic triplets. The result is that the top five most synergistic triplets remain unchanged before and after the bias correction. When looking at the overall composition of the top 50 remaining synergistic triplets, we find that 36 of the triplets are present both before and after the bias correction (see Appendix E). Therefore, the bias correction introduces 14 new synergistic triplets which can be analyzed. None of the most synergistic triplets contain the depression score.

The most synergistic remaining triplet both before and after bias correction is [“UnSatDeg”: Estimated degree of unsaturation, “SLDLTG”: Triglycerides in small LDL; mmol/L, “FAw3”: Omega-3 fatty acids; mmol/L], with $\hat{\Omega }=-0.691$ and ${\hat{\Omega }}_{BC}=-0.592$.

## 6. Discussion

### 6.1. The Naive O-Information Estimation in Simulations

The mean naive O-information estimation $\bar{\hat{\Omega }}$ over 30 trials and their standard deviations are shown in Figure 1 and Figure 3, respectively. The $\bar{\hat{\Omega }}$ values of the fully redundant triplet and the fully synergistic triplet behave as expected, as they seem to follow their theoretical equations $\Omega_{max}=\log K$ and $\Omega_{min}=-\log K$. The slight deviation from the theoretical $\Omega_{min}$ in the synergistic triplet is due to the bias, discussed in depth in the following sections. The $\bar{\hat{\Omega }}$ of the independent triplet very clearly shows the presence of a bias, as we see $\bar{\hat{\Omega }}<0$ for a large section of the heatmap, specifically where N/K is small.

The standard deviations of $\bar{\hat{\Omega }}$ in Figure 3 also align with our expectations. As $X_{1}=X_{2}=X_{3}$ in the redundant triplet, the joint distributions are identical to the marginal distributions. Moreover, the quantile binning strategy ensures that we have a near perfect uniform marginal distribution in every trial. Together, this causes the standard deviation of $\bar{\hat{\Omega }}$ to be approximately zero. The standard deviations for the synergistic and independent triplet are larger, as there two independent variables in the synergistic triplet, and three independent variables in the independent triplet. However, overall the standard deviations are still quite low.

### 6.2. Bias of the Naive O-Information Estimation in Simulations

Before discussing the behavior of the bias δ, we point out that the quantile binning strategy results in the naive entropy estimation $\hat{H}\left(X\right)$ to have zero bias if NmodK=0 and N≥K. Even if NmodK≠0, the bias of $\hat{H}\left(X\right)$ is still very close to zero. This happens because the bins for each variable are constructed based on the *observed* samples (even though we know the ground truth quantiles of the simulated variables), resulting in their quasi-uniform distributions. We do this to make the simulation comparable to the procedure of binning empirical data, for which the ground truth quantiles are *not* known. If a different binning strategy is used, we can also expect the entropy estimation to become biased, thus increasing the bias of the naive O-information estimation. However, it is not trivial to see how the bias in the entropy estimation would affect the bias of the joint entropy estimation.

#### 6.2.1. Bias in the Independent Triplet

A strong bias towards synergy is observed when $K^{2}≲N≲K^{3}$. A sample size smaller than $K^{3}$ implies that not all of the jointly possible bins can be observed. Thus, by definition, an accurate estimation of the joint probability distribution is impossible. This leads to a severe underestimation of $\hat{H}\left(X^{n}\right)$. From the equation of the O-information estimation (Equation 3), it is clear that an underestimation of $\hat{H}\left(X^{n}\right)$ biases the O-information downwards, making it appear more synergistic than it is in reality. Moreover, even if the sample size is equal to or just slightly larger than $K^{3}$, it is unlikely that every unique bin combination is observed at least once. Thus, even above the $N=K^{3}$ line, some bias δ>0 remains, which slowly fades to zero as the ratio N/K increases.

We also see a decreased (but still positive) bias δ when $N≲K^{2}$. Here, the sample size is too small to observe all possible bin combinations between two of the three variables. Thus, it is impossible to even estimate $\hat{H}\left(X_{-j}^{n}\right)$ accurately, resulting in a severe underestimation of both $\hat{H}\left(X^{n}\right)$ and $\hat{H}\left(X_{-j}^{n}\right)$. As seen by the opposite sign of these two terms in (Equation 3), their underestimations cancel each other out to some extent. This results in a smaller bias δ, despite the very small sample size, *N*, and large number of bins, *K*.

Clearly, this smaller bias when $N≲K^{2}$ should not be taken as an improvement. In Figure 3, we see a much higher standard deviation of O-information estimations for the case that $N≲K^{2}$. $\hat{H}\left(X^{n}\right)$ and $\hat{H}\left(X_{-j}^{n}\right)$ are both still severely underestimated, and achieving a careful balance between their two biases is not a reliable method of reducing the bias in the naive estimation.

It is important to emphasize that a significant bias can still be observed with relatively large sample sizes. For example, when N=10,000 and K=50, the triplet of independent variables has $\bar{\hat{\Omega }}\approx -3.11$, while its theoretical O-information should be zero. To put this into perspective, the minimum bound of the O-information for a system of n=3 variables and K=50 bins is equal to $\Omega_{min}=-\log\left(50\right)\approx -5.64$. The naive O-information estimation is thus closer to the minimum O-information bound than to its true O-information of zero.

#### 6.2.2. Bias in the Redundant Triplet

Figure 2 shows that the redundant triplet is biased towards synergy when N≲K. Due to the fact that $X_{1}=X_{2}=X_{3}$, we have that $H\left(X^{n}\right)=H\left(X_{-j}^{n}\right)=H\left(X_{j}\right)$. Thus, the strength of the underestimation of all three of these entropy terms are equal in the estimation of the O-information (Equation 3). The underestimations of $\hat{H}\left(X\right)$ and $\hat{H}\left(X_{-j}^{n}\right)$ cancel each other out in the sum in (Equation 3). Therefore, the only underestimation that remains is the underestimation of $\hat{H}\left(X^{n}\right)$. If N<K, then $\hat{H}\left(X^{n}\right)$ by definition must be underestimated, and we see the bias towards synergy in the naive O-information estimation. We also see that this bias decreases to zero very quickly as soon as *N* increases above *K*. As the quantile binning eliminates the bias in the entropy estimation $\hat{H}\left(X\right)$ as long as N≥K, this also eliminates the bias of the joint entropy terms. Thus, the naive O-information estimation becomes much more accurate as soon as *N* increases above *K*.

#### 6.2.3. Bias in the Synergistic Triplet

For the fully synergistic triplet, the minimum sample size which is theoretically needed to accurately estimate the O-information is $N=K^{2}$. However, we see that, even when *N* is larger than $K^{2}$, the bias δ is not zero. Again, this occurs because it is unlikely that each of the $N=K^{2}$ samples observes a unique bin combination of the $K^{2}$ total possible combinations.

The bias is towards redundancy in the synergistic triplet due to the balance of $\hat{H}\left(X^{n}\right)$ and $\hat{H}\left(X_{-j}^{n}\right)$ in the naive O-information estimation (Equation 3). The joint entropy $\hat{H}\left(X^{n}\right)$ appears once, adding to the O-information estimation, while $\hat{H}\left(X_{-j}^{n}\right)$ appears three times, subtracting from the O-information estimation. In the synergistic triplet, the magnitude of the underestimation of both $\hat{H}\left(X^{n}\right)$ and $\hat{H}\left(X_{-j}^{n}\right)$ are the same per sample size *N*. Thus, the net effect on the bias term is negative, and the effect on the resulting O-information estimation is positive. Therefore, the balance of their underestimations make the naive O-information estimation biased towards redundancy.

Moreover, we do not see the O-information bias fade to zero as fast as in the redundant triplet. This occurs because the quantile binning only eliminates the bias in the entropy estimation $\hat{H}\left(X\right)$, and unlike the redundant triplet, the joint entropy terms are not equal to the marginal entropy terms any more.

### 6.3. The O-Information Bias Approximation in Simulations

In this section, the accuracy of the mean naive estimation $\bar{\hat{\Omega }}$ is compared to the accuracy of the bias-corrected estimation $\bar{{\hat{\Omega }}_{BC^{\prime }}}$. This is performed through the $\varepsilon^{\prime }$ term introduced in Equation (Equation 16). Recall that, if $\varepsilon^{\prime }<0$, $\bar{{\hat{\Omega }}_{BC^{\prime }}}$ is more accurate than $\bar{\hat{\Omega }}$, and vice versa if $\varepsilon^{\prime }>0$. The $\varepsilon^{\prime }$ term for each (N,K) combination is shown in Figure 4. Figure 5 and Figure 6 show the behavior of $\bar{\hat{\Omega }}$ and $\bar{{\hat{\Omega }}_{BC^{\prime }}}$ for the slices shown in Figure 4, where K=10 and K=30. These two figures are helpful to visualize many of the behaviors discussed in this section.

#### 6.3.1. Bias Approximation of the Independent Triplet

The sections in Figure 4 of negative $\varepsilon^{\prime }$ values for large enough *N* are expected. Here, the bias correction partially corrects for the underestimation of the entropy terms in the O-information estimation, which provides a more accurate estimation of the O-information.

We also see that $\varepsilon^{\prime }$ is positive for small N/K ratios, implying that the naive estimation is more accurate than the bias-corrected one. As discussed in Section 6.2, the mean bias of the naive estimation actually decreases when $N≲K^{2}$ due to the canceling out of underestimations of various entropy terms. This effect is not captured by the bias approximation (Equation 12), where a smaller *N* always results in a larger bias approximation. The bias approximation is not able to capture this effect because we use the second-order expansion of the Taylor approximation, instead of the third-order expansion, as given by [31]. The second-order expansion works well for uniform distributions; however, in the independent triplet, it is not guaranteed that the joint probability distributions $H(X^{n})$ and $H(X_{-j}^{n})$ are uniform. Therefore, the balance of biases in $\hat{H}\left(X^{n}\right)$ and $\hat{H}\left(X_{-j}^{n}\right)$ terms results in this overestimation of the bias we observe.

#### 6.3.2. Bias Approximation of the Redundant Triplet

Recall that, in the fully redundant system, we only need to concern ourselves with the bias in the entropy estimations $\hat{H}\left(X\right)$. In the section with a slightly larger *N* but where N<K, the entropy underestimations are partially corrected for. From the O-information estimation (Equation 3), we see that increasing $\hat{H}\left(X\right)$ has the effect of increasing the O-information estimation as a whole to reflect the underlying redundancy.

Around the N=K line, however, the naive estimation is more accurate than the bias-corrected estimation. This is due to the quantile binning strategy used. The quantile binning results in almost perfectly uniform marginal distributions as long as N≥K. Moreover, due to the system constraint that X1=X2=X3, the marginal distributions are equal to the joint distributions. Thus, the naive estimation $\hat{\Omega }$ has approximately zero bias in the fully redundant case. The bias correction, however, is derived under slightly different assumptions than the quantile binning (discussed in more depth in Section 6.4), and therefore tries to correct for a nonexistent bias in the entropy estimation $\hat{H}\left(X\right)$. In the redundant triplet, this results the an overshoot of $\bar{{\hat{\Omega }}_{BC^{\prime }}}$ over the true Ω.

The white dots along the N=K line in Figure 4 are the points where $\bar{{\hat{\Omega }}_{BC^{\prime }}}$ have overshot the true O-information by the same distance as $\bar{\hat{\Omega }}$ has yet to approach the true O-information. Therefore, their absolute errors cancel each other out, resulting in $\varepsilon^{\prime }=0$.

#### 6.3.3. Bias Approximation of the Synergistic Triplet

The bias correction works much better for the fully synergistic triplet than the independent triplet when $N≲K^{2}$. For the synergistic triplet, the bias approximation formula (Equation 12) reduces to the concave parabola $\delta_{MM}=\frac{1}{2Nln(2)}\left[-2K^{2}+3K-1\right]$. This parabola is centered around approximately zero and is very wide, as the $\frac{1}{2Nln(2)}$ coefficient is so close to zero. The larger the sample size *N*, the wider the parabola becomes, and thus a higher *K* is needed in order for the bias approximation to noticeably deviate from zero. Only then can the bias correction take effect for the synergistic triplet. This explains why the bias approximation works particularly well for $N≲K^{2}$, and does not have a noticeable effect when *N* is larger.

### 6.4. A Note on the Quantile Binning Strategy and the Bias Approximation

While the experiments on the simulated data show that the bias-correction process is far from perfect, we point out that our procedure of bin allocation is not completely in line with the idea behind which the O-information bias correction (Equation 12) was derived. As stated above, the bins are constructed based on the observed samples, even though we know the ground truth quantiles of the generated variables. However, the bias-correction process (Equation 12) assumes that the bins are created from the true underlying distribution, and the observed samples are assigned the bins they happen to fall into.

This is illustrated by the fact that in the fully redundant triplet when NmodK=0, the naive O-information estimation has zero error while the bias-correction process does not have zero error, and tries to correct for the underestimation in the entropy we would see if the samples were assigned to the bins constructed from the true underlying distribution.

Therefore, the bias correction might have performed better in this simulated experiment if we had constructed the bins from the known underlying distribution, and assigned the observed samples to these existing bins. As this is impossible with empirical data, we decided to keep the binning procedure of the simulation experiment comparable to the empirical data.

### 6.5. When Can the Bias-Correction Process Be Applied to Empirical Data?

The redundant triplet shows next to no bias for sample sizes *N* that are larger than the number of bins, *K*, to begin with; while the bias correction is less accurate than the naive estimation around N=K, this inaccuracy fades to zero relatively quickly as N/K increases further. As it is generally recommended to have a sample size much larger than the number of bins, it is unlikely that the errors of the bias correction in redundant triplets pose any serious problems when applying this bias correction. The situation is more tricky for triplets of independent variables. If the sample size is $N≲K^{2}$, then the bias correction does more harm than good. However, despite the naive estimation performing better in this case, it is still quite biased itself. The bias correction only improves the O-information estimation for triplets of independent variables given that $N\gg K^{2}$. However, it only partially corrects for the bias. If the triplet is synergistic, the bias correction works very well for $N≲K^{2}$, substantially improving the O-information estimation. If $N≳K^{2}$, then there is very little bias to begin with, and the bias correction can only very slightly improve the estimation further.

Assuming we have no prior knowledge about the joint distribution of variables, the prudent approach is to assume independence between variables in a triplet, as these triplets show the largest error of both the naive estimation δ and the bias-corrected estimation $\delta_{BC^{\prime }}$. With this assumption, the bias-correction process can be applied only if $N\gg K^{2}$, but it is important to keep in mind that it does not correct the bias fully. If $N≳K^{3}$, the naive O-information estimation should quite good to begin with, and the bias correction would simply improve it slightly more.

### 6.6. Application to the Young Finns Study

The simulated independent triplets have a mean O-information estimation (both naive and bias-corrected) that aligns almost perfectly with the peak of the O-information distribution of the empirical data. This reveals that a large majority of triplets in the YFS dataset are independent from each other (or their synergy and redundancy cancel each other out perfectly), rather than synergistic as their negative O-information might imply.

The result that almost all of the top synergistic triplets are ratios makes sense as well. For triplets which are exact ratios, the pairwise constraints are much weaker than the constraints on the system, which is completely governed by the equation X3 = X1/X2. This perfectly aligns with the idea that a synergistic system has local independence but global cohesion. Two of the three variables in the triplet must be known in order to derive the third, but just knowing one does not give much information about either of the other two variables. Even though the ratio triplets in the YFS dataset are not exact ratios due to the discretization for example, they were still identified as highly synergistic. This provides a proof of concept of the O-information, as it indeed assigns the most negative O-information to the systems in empirical data which embody this concept of synergy.

After removing the triplets containing inherent mathematical relationships (ratios, sums, and means), the fact that the bias correction does not change the order of the top 5 synergistic triplets implies that there is a level of robustness in the O-information. It suggests that the strength of the bias does not drastically change across similarly synergistic triplets. This allows us to make interpretations (with caution) about the relative strength of higher-order synergistic relationships. It also allows us to single out specific variables that frequently occur across highly synergistic triplets, with the reasonable assumption that these variables remain relevant after correcting for the bias. However, the bias correction does change the ordering beyond the top 5 synergistic triplets, and introduces 12 new triplets to the 50 most synergistic ranking. Thus, in combination with the simulations, the bias correction could provide valuable insights into synergistic triplets that may previously have been overlooked, for instance when performing centrality analyses on this hyper-network.

The most synergistic remaining triplet both before and after bias correction is [“UnSatDeg”: Estimated degree of unsaturation, “SLDLTG”: Triglycerides in small LDL; mmol/L, “FAw3”: Omega-3 fatty acids; mmol/L], with $\hat{\Omega }=-0.691$ and ${\hat{\Omega }}_{BC}=-0.592$. The degree of unsaturation is determined by the total number of double bonds and rings present in molecules such as triglycerides. The number of double bonds present in a triglyceride is in turn influenced by its fatty acid composition [32]. Omega-3 fatty acids are a type of polyunsaturated fatty acid which can be found in triglycerides [33]. The “poly” signifies the existence of multiple double bonds, and each of these double bonds increases the overall degree of unsaturation of the triglyceride. The third variable SLDLTG represents the triglyceride content in small LDL particles. A potential explanation for the synergy between these three variables is that the degree of unsaturation cannot be determined by the amount of Omega-3 fatty acids or the amount of triglycerides in LDLs alone. However, together they provide more information on the overall composition of saturated and unsaturated fatty acids in LDLs, and thus the degree of unsaturation can be more accurately estimated.

## 7. Conclusions

An important finding is that in triplets of independent variables the bias in the naive O-information estimation δ can persist even with relatively large sample sizes. In contrast, the O-information estimation of fully redundant and synergistic triplets is less biased, as their joint probability distributions can be estimated with less samples.

For systems of n=3 independent variables, the naive O-information estimation is severely biased towards synergy if $K^{2}≲N≲K^{3}$. At this sample size, especially the joint entropy $H(X^{n})$ is severely underestimated, resulting in a strong downward bias of the naive O-information estimation. Without taking this bias into account, higher-order relationships may be classified as highly synergistic while in reality being close to independent. In fact, our simulations show that, even with a larger sample size of N=10,000 and K=50, the naive O-information estimation is so biased that it is closer to the minimum bound of the O-information than its true value of zero.

This implies that highly synergistic groups of variables identified in empirical data using the O-information may in reality be closer to independent, if the N/K ratio is small enough and no bias correction attempts have been made. Indeed, findings illustrated by the Young Finns Study and our simulations suggest that a large majority of triplets in the dataset which would naively be labeled as synergistic due to their negative O-information are actually independent. When looking for synergistic higher-order relationships in empirical data, it is therefore crucial to be aware of the existence of this bias. It may also be helpful to use simulations to establish the O-information estimation value which fully independent triplets with the same *N* and *K* values would have, in order to benchmark the boundary between synergistic and redundant in the empirical dataset.

Future work could improve the Miller–Maddow bias-correction process derived in this paper by exploring better ways to estimate $K^{-j}$ and $K^{\left(n\right)}$ from the data, rather than simply counting the number of joint bin combinations we observe. Moreover, other O-information bias correction methods should be explored. The Jackknife method may be a better alternative than the Miller–Maddow method for small sample sizes, if one is willing to take on the additional computational cost. Other alternatives with lower computational cost should also be explored, such as Harris’ third-order Taylor approximation or Bayesian approaches.

Moreover, future work should explore the effect of different discretization methods on $\hat{\Omega }$, δ, and ${\hat{\Omega }}_{BC^{\prime }}$, in order to apply these results to cases where quantile binning may not be appropriate. While it is out of scope for this paper to compare the simulation results for various discretization methods, it is certainly important to keep in mind that the discretization method will affect the behavior of information theoretic measures [34], including $\hat{\Omega }$, δ, and ${\hat{\Omega }}_{BC^{\prime }}$.

Different discretization methods also open the door to exploring the estimation bias δ when variables have marginal distributions other than uniform. For instance, in the case of gaussian marginal distributions, the inherent relationship between the entropy and the variance may be exploited to estimate the O-information, following the method of [18]. While this would still require the estimation of the variables’ variances and the system’s covariance matrices, it would provide an interesting comparison to the bias correction presented in this paper.

Lastly, the analyses carried out in this paper for systems of n=3 could be extended to consider systems of more than three variables. The O-information’s power lies in its scalability to systems of higher dimensions, which should be taken advantage of, while the derived O-information bias approximation term is flexible to include a higher number of variables, it would be important to explore the effect of increasing the number of variables on the approximation’s accuracy. 

## Figures and Tables

**Figure 1 entropy-26-00837-f001:**
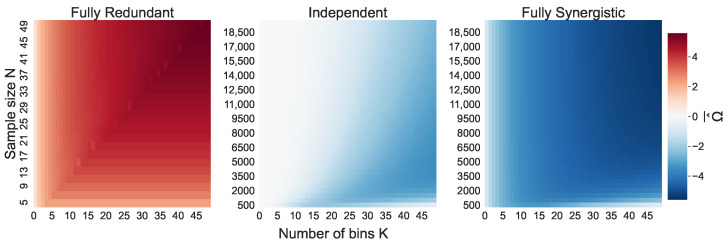
The mean naive O-information estimation $\bar{\hat{\Omega }}$ per (N,K) combination over the 30 trials. This analysis is performed for the fully redundant triplet (**left** panel), the triplet of independent variables (**middle** panel), and the fully synergistic triplet (**right** panel).

**Figure 2 entropy-26-00837-f002:**
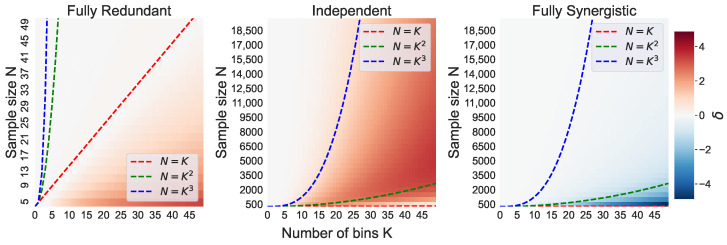
The bias δ of the naive O-information estimation per (N,K) combination. Boundary lines indicate the theoretically minimum sample size, *N*, needed to observe the number of bins, *K*, as well as the joint bin combinations $K^{-j}$ and $K^{\left(n\right)}$.

**Figure 3 entropy-26-00837-f003:**
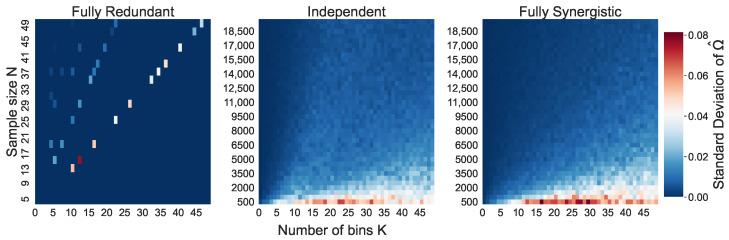
The standard deviation of the naive O-information estimations ${\hat{\Omega }}_{1},\ldots ,{\hat{\Omega }}_{30}$ per (N,K) combination over the 30 trials, for each of the three simulated systems.

**Figure 4 entropy-26-00837-f004:**
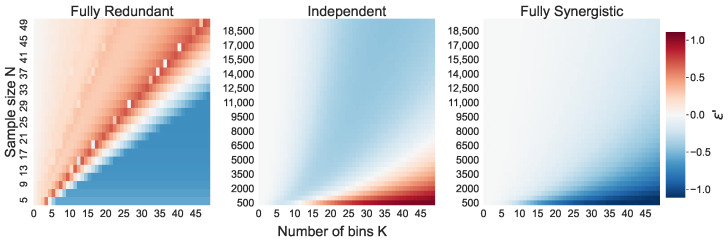
The difference in an estimation error between the naive and bias-corrected O-information estimation ($\varepsilon^{\prime }$). A negative value (blue) implies that $\bar{{\hat{\Omega }}_{BC^{\prime }}}$ is more accurate than the naive estimation $\bar{\hat{\Omega }}$. A positive value (red) implies that the naive estimation is more accurate.

**Figure 5 entropy-26-00837-f005:**
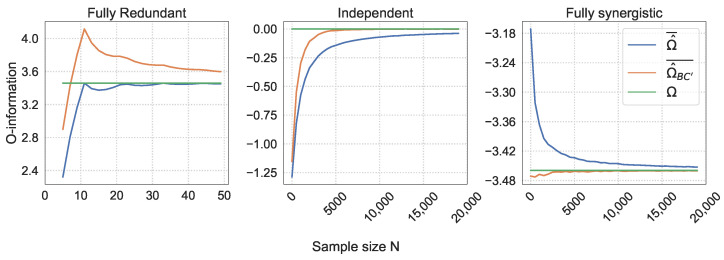
Behavior of the true O-information Ω, the mean naive O-information estimation $\bar{\hat{\Omega }}$, and the bias-corrected O-information estimation $\bar{{\hat{\Omega }}_{BC^{\prime }}}$ for K=10 bins.

**Figure 6 entropy-26-00837-f006:**
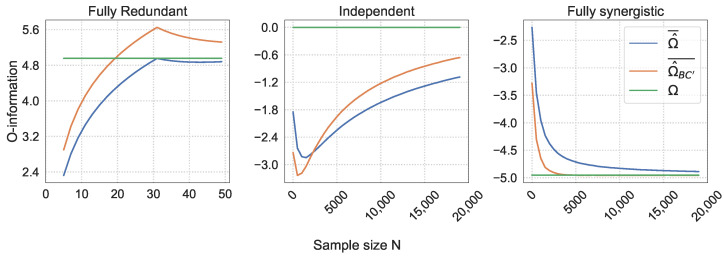
Behavior of the true O-information Ω, the mean naive O-information estimation $\bar{\hat{\Omega }}$, and the mean bias-corrected O-information estimation $\bar{{\hat{\Omega }}_{BC^{\prime }}}$ for K=30 bins.

**Figure 7 entropy-26-00837-f007:**
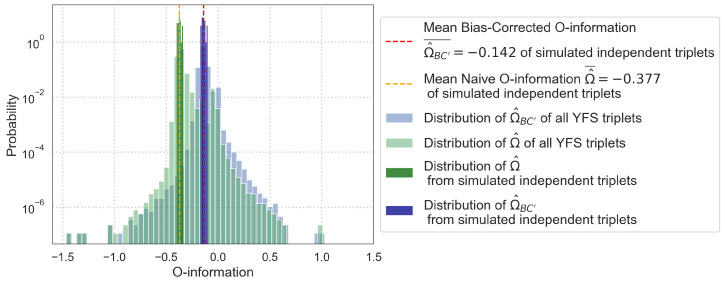
Distribution of the naive estimated O-information $\hat{\Omega }$ and the bias-corrected estimated O-information ${\hat{\Omega }}_{BC^{\prime }}$ for all triplets in the feature selected dataset. The dashed lines of $\bar{\hat{\Omega }}=-0.377$ and $\bar{{\hat{\Omega }}_{BC^{\prime }}}=-0.142$, respectively, indicate the mean naive O-information estimation and the mean bias-corrected O-information estimation of the simulated independent triplets. The darker shaded distributions around the dashed lines are the distributions of the simulated O-information estimations.

## Data Availability

The GitHub repository to run the simulations can be found at https://github.com/johannagehlen/oinfo-bias (accessed on 5 July 2024) and at Zenodo with DOI 10.5281/zenodo.13760381. Restrictions apply to the availability of the Young Finns Study data. Data were obtained from the Research Centre of Applied and Preventive Cardiovascular Medicine and Centre for Population Health Research, University of Turku.

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
