# Peer review of "Bias in O-Information Estimation"

_entropy, 2024, doi:10.3390/e26100837_

Round 1

Reviewer 1 Report

Comments and Suggestions for Authors

The authors study how sample size affects the bias of the O-information, a measure that estimates the balance between synergy and redundancy in a system. They provide a nice and concise introduction to O-information, derive Miller-Maddow-type approximation for the bias of the O-information in binned data, and study how the bias changes with sample size and number of bins in a variety of settings, including empirical data. I really enjoyed reading the paper, which is generally well-written and very detailed. All the background information is present, and the results are clearly explained. Overall, I believe that the manuscript stands as a technically sound contribution to the literature. I have a few minor comments that I believe the author should address before the paper can be published.

1) The authors should consider discussing the fact that, in general, estimating the joint entropy distribution from samples of high-dimensional continuous variables is an incredibly challenging task, as I am sure they are aware. Standard estimators based on k-nearest neighbors distance fail rather badly in high-dimensional settings, and the subject is an active topic of research (see, e.g., C. Lu and J. Peltonen, 2020 for a recent correction to improve the estimation in high dimension, or P. Czyz et al, NeurIPS 2023 for the effects on mutual information estimation). Thus, I am not sure I understand how the joint entropy term in Eq. (1) does not quickly become intractable due to the curse of dimensionality, from which all estimators will suffer. Similarly, I am not sure I understand when they write in line 66 that O-information can be easily estimated from continuous, non-binned data exactly for this reason. I believe it would be fitting to add a discussion along these lines, as the dimensionality of the system should play a rather important role.

2) The authors mention in several places that "in synergistic systems, variables tend to be pairwise independent". While I believe I understand what they mean, and the example of the synergistic triplet makes it clear, I think that the wording may confuse the readers. Unless I am missing something, if in a system all variables are pairwise independent, no synergy or dependency at all can be present. I would rephrase this part to avoid confusion in the readers unfamiliar with the subject.

3) The authors study the case of a synergistic triplet, where X1 and X2 are independent, and X3 depends on X1 and X2 both. Along these lines, other classes of physical systems may play a similar role, where variables are instead conditionally independent, but depend on an external "environmental" variable. A typical example is stochastic processes in random media or environments, see Chechkin et al, PRX 2017, W. Wang et al, Journal of Physics A 2020, or Nicoletti and Busiello, PRL 2021. For three variables, these cases would be more similar to the following: X1 and X2 depend both on X3, but do not directly interact with one another, so that the joint probability of the system would factorize as p(X3)p(X1|X3)p(X2|X3). What can be said about conditionally independent systems such as these ones? Are there by any chance more similar to the "almost" independent case mentioned in the previous point? Since these cases have direct physical applications, a discussion along these lines may be interesting to a more general readership.

4) In section 4.3, the acronym "CVD" is introduced, but it is not specified what it means.

5) I appreciate that the author shared a GitHub repository to replicate the results. Would it be possible to deposit the code in a more permanent repository, such as Zenodo, for future accessibility?

Reviewer 2 Report

Comments and Suggestions for Authors

the revisions are attached 

Comments on the Quality of English Language

N/A

Round 2

Reviewer 2 Report

Comments and Suggestions for Authors

The authors have addressed all my comments. The paper is solid and relevant to the topic of estimating higher order interactions.

I recommend the publication of this paper.